# Examining Risk Absorption Capacity as a Mediating Factor in the Relationship between Cognition and Neuroplasticity in Investors in Investment Decision Making

**Yadav Devi Prasad Behera** [1] , **Sudhansu Sekhar Nanda** [2,*] , **Shibani Sharma** [3] **and Tushar Ranjan Sahoo** [4]

[1] Department of Business Management, Central University of Odisha, Koraput 763004, Odisha, India; deviprasadyadav2009@gmail.com
[2] Kirloskar Institute of Advanced Management Studies, Harihar 577601, Karnataka, India
[3] Department of Business Administration, Gangadhar Meher University, Sambalpur 768001, Odisha, India; shibanisharma1947@gmail.com
[4] National Institute of Science and Technology, Berhampur 761008, Odisha, India; sahoo.tushar571@gmail.com
[*] Correspondence: nandasudhansusekhar.87@gmail.com

**Abstract:** The encouragement of potential investors who are emotionally broken by past losses and market experiences is crucial to the sustainable flow of funds to the stock market. This can be established by building a knowledge-creating mechanism among investors in their cognitive dimensions, which, in turn, can develop their risk-bearing potential to reach the optimum level so that emotionally broken investors can use their cognitive abilities with their developed risk-absorption potential to further invest in the market in the near future. This study investigates the mediating effect of risk-absorption attitudes in the relationship between cognition and neuroplasticity in investors. Data for the study collected from 506 individual retail investors' samples using a stratified random sampling technique were analyzed through covariance-based structural equation modeling. The findings of the study indicate that the constructs, viz., the investors' cognition, risk absorption, and neuroplasticity, are valid and reliable. The structural model also supports the notion that risk absorption mediates the relationship between the investors' cognition and neuroplasticity. The outcomes of the study are expected to aid in the policy formulation for equity-related financial product marketers, such as depository participants, brokers, mutual funds and SIP institutions, and to help in healing psychological trauma that potential investors suffered from due to losses in the past and overcoming reluctances to further invest in stock markets. The investors' terrible psychological health developed because of past loss experience can be restored through the concept of neuroplasticity, in which different cognitive dimensions are used, while also enhancing risk absorption in potential investors.

**Keywords:** risk absorption; risk tolerance; neuroplasticity; cognition; behavioral finance; CB-SEM

**JEL Classification:** E71; G41; G24

## 1. Introduction

A sustainable economy always requires sustainable financial power, but a crisis in world economy significantly affects India's economy (Himanshu et al. 2021; Liu et al. 2020), as it mainly depends on major foreign direct, portfolio and institutional investments for its financial needs (Long et al. 2017). The investment from domestic individual investors is only 3.7% of the total investment pool, in comparison to 55% in the U.S.A. (Balwani et al. 2021). These statistics show that the flow of funds to Indian securities markets is the result of the contribution from external sources rather than domestic sources, which makes the Indian economy vulnerable to external crises (Hoffmann et al. 2010). Therefore, it is necessary to inject more funds from domestic sources to make the economy sustainable in the future.

The existing available literature focuses on the decision making or the attitude that affects the investment decisions of individual investors. However, little of the literature focuses on encouraging potential and emotional investors, those mainly found in the Indian subcontinent. The reasons behind investment decisions are mainly studied by many researchers, but the encouragement of emotionally driven investors and investors with negative experiences, or experiences of investment loss, have not been taken into consideration.

Indian equity markets faced several scams in the past, which injured the memory and psychology of the investors, creating a negative perception towards the market (Mehta 2019). Many investors consider the act of investing in the stock market as gambling, as the sudden fall without sufficient relevant information makes the stock market a vulnerable rather than trustworthy investment avenue (Vijayakumar 2021). It was due to the neural or psychological injury occurred because of the loss in investment due to uncertainty in the market, which makes investors to invest further in same risky market (Bhakta 2017). With this psychological breakdown in relation to the prospect of stock market investments, one cannot also hope for a sustainable economy. Domestic investment is crucial for economic growth, and with the investors' psychological, memory or neural injuries, it has been very difficult to manage the economy with the aim of sustainability (Mishra et al. 2020; Saleem and Zaheer 2018). Therefore, it is imperative to understand the psychology of investors, those who have borne the loss in the past and donot intend to invest any further, and to motivate them to invest again through proper planning and information processing (Hoque et al. 2018; Pless et al. 2017). The current study classified this act as the 'neuroplasticity' of the investors, with the intention to repair the ill-functioning of the investors' intellect. This study can develop reasoning capacity to the investors to recover from past mental trauma, so that they can make further investments in the future after obtaining a proper education and understanding in the field of investment (Rugnetta 2020).

The development of cognitive abilities among the investors is vital so that neuroplasticity can alleviate the negative memories of the past and build a reasoning and calculative way to invest (Kaur and Vohra 2012). The different dimensions of cognition are the outcomes of the different mental processes, which use different sources of data to produce relevant information for a particular objective (Brandimonte et al. 2006). The dimensions may be cold, hot, social and meta (Niznikiewicz 2013). Cold cognition is mental processing that uses information from reliable sources and analyzes it for developing investment decisions (Bhushan 2014; Tully and Niendam 2014). Hot cognition prefers the sources of information gathered from acquaintances and experts, and analyzes it to research the conclusion (Bechara 2004). Social cognition follows the scenario prevailing in society and evaluates the possible outcomes benefitting the investor (Carol 2003; Mcdonald 2013). Finally, the metacognitive element learns from experience and past data and speculates on the future based on past knowledge and applicability (Hosaini and Saadatmand 2014; Kumaravelu 2019).

The current study has not only focused on the reasons behind investment decisions and the effect of negative experiences on future decision making, but also emphasizes the different way of developing cognitive abilities in investors by providing information through different sources so that relevant information will be accessible by potential investors. After getting information, the investors can judge the reasons behind their past mistakes in investment and can think of the necessary actions to be taken in the future to avert losses and to obtain good targeted profit.

It also helps to ascertain the strength and weaknesses of any past incident or decision. Apart from the cognitive process, risk adherence can also help in neuroplasticity in the long run (Rustichini 2005). In the presence of knowledge and with the proper processing of information, the risk-tolerance attitude can be enhanced to the extent of investing for longer durations of time as a repetitive investment (Bezzina et al. 2014; Sheedy and Lubojanski 2018). In this context, the present research question is "Can cognition facilitates neuroplasticity in investment behavior with a mediating effect from risk absorption?" The objectives framed for the research are:

- To investigate the effect of different dimensions of investors' cognition (ICO) on neuroplasticity in investors (NPL).
- To explore the effect of different dimensions of investors' risk absorption (RAB) on neuroplasticity in investors (NPL).
- To evaluate the effect of different dimensions of investors' cognition on investors' risk absorption capacity.

The present study focuses on the systematic literature review to find a model for the deductive approach of research. The model is constructed based on the different relations between the observed variable related to the field of management, psychology and medical science elements. The validity and reliability of the model being tested through structural equation modeling techniques and the suggestions are presented based on the proved model.

## 2. Literature Review

Learning predominantly affects the attitude of any human being and this learning is persuaded by three basic elements, viz., the cognitive element, affective elements and psychomotor elements (Hoque 2016; Krueger et al. 2015). Investment attitudes relating to the stock market are mainly influenced by the knowledge processing capacity of the investors, i.e., the cognitive elements and the level of risk absorption (Brandimonte et al. 2006). The cognitive perception is again classified on the basis of the sources of information, as well as the progression of information management by the human brain based on the ability to reason (Vitor et al. 2019).

### 2.1. Dimensions of Investors' Cognition

Cognition is the human activity of learning, acquiring knowledge, processing knowledge and the use of such knowledge for specified activities (Tidwell et al. 2000). Cognition affects the awareness, perception, judgement and reasoning of an individual, which eventually arises from the information sources and information processing ability (Frijns and Indriawan 2018). In this regard, cognition is classified as cold cognition, hot cognition, metacognition and social cognition (Hoque 2016). The segment of the cognitive process that enumerates the non-emotional style of information processing and use such information for the pre-stated purpose is identified as cold cognition (Robbins 2011). A person uses a sensory organ to collect relevant information through retrospective learning; moreover, the memory plays its part in subsequent activities (Statman 2017). An investor, being a rational creature, uses his or her sensory organ to collect information from various sources, viz., the company's website, newspaper articles and publications, sharing fundamental information relating to the company required for any equity market investment decisions (David and Matu 2017; Ferree and Merrill 2000). Analyzing financial indexes concerning economic, political and psychological indicators helps to make better investment decisions (Tooranloo et al. 2020). Where cold cognition symbolizes the acceptance of relevant information by the self and processing it with an unemotional point of view, hot cognition, on the other hand, is affected by attention bias (Aggarwal 2019), which is an emotional way of processing information by prioritizing a certain type of stimuli over others (Thagard 2006). In this case, the prioritized stimuli are considered to be the information that is obtained from specific people, such as experts, family members and peer groups (López-cabarcos et al. 2019). Hot cognition is basically considered as the information that is obtained from known people (Healey et al. 2018), and is processed according to the required benefits (Xie and Yang 2015). Investors in India are usually emotional and seek those sources of information that come from the peer (David et al. 2002; Palmer 2015) or known people, and use that information after special consideration for investments (Ask and Granhag 2007). Metacognition is considered to be the highest quality of information-processing activity, where a person precisely monitors, plans and assesses self-understanding and performance. Metacognition supports the critical analysis of both self-thinking and learning that comes from all the sources of information. The persons with a metacognitive ability acquire knowledge

from their experiences and awareness. Investors obtain knowledge from their encounters, investigations and past experiences (Talwar et al. 2021). The knowledge derived through metacognitive activity is superior as it tends to follow the experiences that arise from planning and practical applicability (Duong et al. 2014). Metacognition helps to identify the strengths and flaws in order to find the gap in their knowledge. It helps to plan, observe and assess their performance. Uncertainties are avoided with metacognitive innovation, which guarantees a better return from decisions (Sohaib et al. 2019). Cognitive efficacy helps to formulate a proper decision, that can result in a better outcome (Yang 2015). Understanding the market dynamics with good observation, from time to time, helps to find the gap that creates market inefficiencies (Jain et al. 2020). This analytical viewpoint helps to produce a better investment avenue (Sharma and Kumar 2020). Metacognition considers an analytical way of working that helps in problem solving (Lee et al. 2013)

Social cognition is proven to be that sort of mental activity that is concerned with concentrating, remembering and thinking about the information that comes from others in the social world (Syriopoulos and Bakos 2019), and creates perceptions through information processing ability (Fiske 1993). An investor, being a human being, is a part of society and comes across different individuals who can influence and affect the decision-making power (Haritha and Uchil 2020). In this circumstances, the investors collect pertinent data, which can be beneficial for investment yields from society at large and process it for further benefits. The society at large signifies the area in which it resides, the people around whom it communicates in daily life and, moreover, the social interactive platforms, in which they interact. Even social–cultural knowledge significantly affects the decision-making process relating to investments (Bondia et al. 2019).

### 2.2. Cognition and Neuroplasticity in Investors

Knowledge has been the answer to all problems and the knowledge derived from cognitive processes can cure all the psychological disruptions created as a result of negative experiences (Njegovanović 2018). Often, investors fear leads to panic due to market crashes (Mushinada 2020), but cognitive process results in the realization of the relevance of information and helps to cope with self-hurt behavior that is the result of past losses (Erkut et al. 2018). A proper education develops prior knowledge that leads to financial literacy. This financial literacy helps one to understand the reason behind a past decision resulting in loss and helps to create proper strategies for the future, resulting in neuroplasticity (Frydman and Camerer 2016). Financial literacy developed due to different cognitive factors, which lead to continuous market participation (Sivaramakrishnan et al. 2017). The above arguments illustrate Hypothesis 1.

### 2.3. Dimensions of Risk Absorption

Investment risk varies in different investment avenues and the capacity of bearing this investment-related risk also varies among different investors (Anantharajan and Sachithanantham 2016). Investors are generally classified into three categories based on their risk-tolerating capacity: risk takers, risk neutral and risk averters (Chen 2020; Outreville 2014). However, risk-taker or risk-tolerant investors have a limit of being able to tolerate risks for a specific duration of time. Investors having an extended risk-bearing nature (Díaz and Esparcia 2019) and preferring to make risky investments for a longer duration of time are presumed to have the attitude of risk absorption (Behera et al. 2021). Investors with a greater veracity tend to make continuous investments in similar risky financial products, which show risk-absorbing behavior. The risk-absorbing investors are considered to be those who intend to bear greater risks every time they invest in the equity market to obtain a greater return from investment (Vasileiou 2021). Investors with a risk-absorption capacity seek more risky investments as they can evaluate the financial risks associated with the investment for different criteria with the help of their cognitive ability (Carr 2014). Planning is an integral part of any investment decision. Mere speculation may not be able to justify the return that investors intend to obtain (Akhtar and Das 2019). Risk-absorbed investors

plan and evaluate the investment avenue before making an investment. In the case of a decline in the value of the invested securities, they wait for the market value to increase, instead of selling the shares right away (Kannadhasan 2015). This confidence in investments arises only as a result of learned knowledge and detailed information processing. Financial literacy helps to assess the investment avenues based on risk propensity to produce proper planning before making an investment decision (Aydemir and Aren 2017). Strategic planning helps to diversify the investment so that investment risks may be avoided, which ultimately helps in taking more calculative risks in investment (Tang et al. 2008).

## 2.4. Risk Absorption and Neuroplasticity

The antecedent of risk absorption is said to be the behavioral components that instigate the risk-absorption attitude, which is the recurring behavior of investing in similar investments and for a longer duration, the risk-seeking behavior that induces the investors to invest in more risky avenues, irrespective of speculation and strategic planning after processing the required information This shows that the character of strategic planning behavior is essential for taking an interest in an investment by forgetting past losses (Behera et al. 2018). Investors with risk-absorption behavior make repetitive decisions to repurchase a similar product. This behavior shows their recurring investment nature (Kannadhasan 2015). With greater consideration of the psychological risks of an investor, the purchase intention and repurchase decision are boosted. Strategic planning helps to diversify the investment so that investment risks can be avoided, which ultimately helps in taking more calculated risks in the future, showing the neuroplastic nature (Tang et al. 2008). In this context, research Hypothesis 2 has been drawn.

## 2.5. Cognition and Risk Absorption

Knowledge developed through cognitive activity has encouraged human beings to take certain calculated risks (Kusev et al. 2017). Relevant information can be produced through proper cognitive sources accessible to the investors that help them to make risky investments and to obtain a better result (Blajer-Gołębiewska et al. 2018). The risk tolerance attitude can be developed by investors' cognitive abilities, which helps in decision making (Khan 2017). Investors' behavior arises from their cognitive abilities through information acquisition and processing can influence the risk-taking capacity that affects stock trading behavior (Tauni et al. 2017). Financial literacy, on the other hand, helps in reducing cognitive biases and brings confidence to the investment (Özen and Ersoy 2019). The above arguments result in Hypothesis 3.

## 3. Research Design and Methodology

The present study followed the philosophy of interpretivism using a deductive approach. Both qualitative and quantitative methods are applied to draw inferences. A causal or causative research strategy is used to find the effect of the independent variable directly on the dependent variable and indirectly through a mediating variable. Cross-sectional data are collected from 506 respondents from 4 strata in India by using a stratified random sampling method. The population considered for the study is the potential investors, who once invested in the stock market and suffered losses, which caused psychological strain. Latent constructs are drawn from the existing literature and the relations are proved from the pre-existing literature of different conceptual and empirical studies. The questionnaire was designed with 20 questions related to cognition, 10 questions related to risk absorption and 3 questions related to neuroplasticity. All 33 questions of the questionnaire are measured based on a 5-point Likert scale. Covariance-based structural equation modeling is applied by using Statistical Package for the Social Science-23v (SPSS-23) and Analysis of Moment Structures (AMOS). The mediating model explains that the independent variable, i.e., Investors' Cognition (IC), directly and indirectly affects the dependent variable, i.e., Neuroplasticity (NPL), through the mediating variable, i.e., Risk Absorption (RA). Model fitting parameters, such as CMIN/DF, GFI, RMR, AGFI, PGFI, TLI, IFI, CFI and RMSEA, are

tested. The structural model is also tested based on rejection criteria. The latent constructs of the measurement models are tested based on construct validity and discriminant validity through scale reliability, composite reliability, average variance explained and maximum shared variance. Common method biases are tested through Herman's one-factor test. Social science research software, such as SPSS v23, is used along with AMOS software for analysis. Literature from sources, such as Web of Science and Scopus, were studied to verify the interrelations among the studied variables. References were made through Mendeley with the 7th APA style of referencing.

### 3.1. Research Hypothesis

**Hypothesis 1 (H1).** *The different dimensions of investors' cognition (ICO) significantly affect the neuroplasticity ofthe investors (NPL).*

**Hypothesis 2 (H2).** *The different dimensions of investors' risk absorption (RAB) capacity significantly affect the neuroplasticity of the investors (NPL).*

**Hypothesis 3 (H3).** *The different dimensions of investors' cognition significantly affect the investors' risk-absorption capacity.*

### 3.2. List of Variable for Measurement Model

The Table 1 illustrates the questions used for data collection and the codes representing them.

**Table 1.** Variables, taken in questionnaire for data collection and their codes for analysis in the measurement and structural model.

| Sl. No. | Variables in the Questionnaire for the Measurement and Structural Model |
|---|---|
| 1 | CC1: Investments in the stock market need knowledge relating to it. |
| 2 | RI1: I have been an active investor in the stock market. |
| 3 | CC2: I read the information on the company's website before investing. |
| 4 | CC3: I read newspaper articles related to investment avenues from time to time. |
| 5 | MC1: I compare similar investment avenues before making a purchase decision. |
| 6 | MC2: I will invest more, if I will receive depth investment training. |
| 7 | RS1: Investing in stock can resolve my greater financial needs. |
| 8 | HC1: I prefer views from experts' stock market investors. |
| 9 | RS2: Stock market investment gives me more income than FD. |
| 10 | HC2: I consult my family members before making an investment decision. |
| 11 | CC4: I collect and read past and expected returns before making an investment. |
| 12 | SC1: I prefer watching morning news on TV and newspapers relating to the same-day investment. |
| 13 | HC3-People around me also gives me suggestions to invest. |
| 14 | MC3-: I prefer an in-depth analysis of a 5-year profit before making an investment. |
| 15 | SP3: Past losses do not stop me from investing. |
| 16 | HC4: Friends and relatives help me to make better stock market investment decisions. |
| 17 | SC2: I prefer to observe the Twitter handles and Facebook pages of big stock market investors. |
| 18 | SC3: Invest mobile apps help me to make better investment decisions. |
| 19 | CC5: -Financial literacy is a must for making investments. |

**Table 1.** *Cont.*

| Sl. No. | Variables in the Questionnaire for the Measurement and Structural Model |
|---|---|
| 20 | SP1: I will continue with the same investment, even if the prices are currently low, if analyzed properly. |
| 21 | MC4: I usually study scholarly articles about stock market investment to obtain greater knowledge. |
| 22 | SC4: The financial advisor in our society helps me to make investments. |
| 23 | MC5: I try to analyze the reasons for the fall and rise of a stock market index. |
| 24 | RI2: I will repeat a similar investment in the future. |
| 25 | NP1: In the future, I can make a better investment decision. |
| 26 | RS3: I want to invest more in comparably risky shares. |
| 27 | RS4: Past investment experiences help me to make more investments. |
| 28 | NP2: I am mentally prepared to handle investment securities in the future. |
| 29 | SP2: I am not emotional, but rather rationally choose my investments. |
| 30 | CC6: Transparency of investment information presents me with more reasons to invest. |
| 31 | RI3: If my income rises, I will make more investments. |
| 32 | MC6: I can make more investments if I have confirmed analytical news. |
| 33 | NP3: I will soon make more investments as my ability has improved. |

*3.3. Sample and Respondent Profile*

Out of the total number of respondents (506), 9 belong to the less than 20 years age group, 264 belong to the age group between 21–30 years, 97 to the age group between 31–40 years, 74 to the 41–50 years, 56 to the 51–60 years and 6 to the age group above 60 years. A total of 348 males and 158 females were used for the study. A total of 293 among the sample were single and 213 were unmarried. A total of 21 persons were below graduation level, 172 were graduation level, 209 were post-graduation level and 104 had another professional degree. A total of 107 persons had a job in government services, and 100 were from the private sector. A total of 207 had their own business and 92 provided professional services. A total of 189 persons had monthly income of less than 20,000, 81 persons had an income range between 20,001–40,000 per month, 115 persons received 40,001–60,000, 71 received an income between the range of 60,001–80,000, 40 between 80,001–100,000 and 10 persons had a monthly income of more than 100,001.

**4. Empirical Analysis and Result**

*4.1. Neuroplasticity Measurement Models*

A measurement model (Figure 1) through the 2nd order CFA (confirmatory factor analysis) technique was tested for the reliability and validity of the constructs, which are the Investors' Cognition (ICO), Risk Absorption (RAB) and Neuroplasticity (NPL) in the measurement model. The measurement model proves the validity and reliability of the above three constructs, and the structural model shows the interrelation among the three constructs as well as the effect of the independent constructs (ICO and RAB) on the dependent construct (NPL). These factors were obtained through the literature review and confirmed through the above-mentioned technique in the context of the neuroplasticity of investors' attitudes. The results are interpreted as follows.

The above values are taken for sufficing the reliability and the validity of the constructs (ICO, RAB and NPL) The reliability is measured through scale reliability and composite reliability. Composite reliability (CR) must be above 0.700 for each factor (Gefen et al. 2000; Hair et al. 2017; Sani et al. 2019). The composite reliabilities of NPL, ICO and RAB are 0.766, 0.809 and 0.786, respectively, which is above 0.700, so the factors are acceptable for the model. The validity of the above-mentioned constructs are measured through AVE and MSV. The average variance extracted (AVE) must be above 0.500 for each factor for model

validation (Almén et al. 2018; Hair et al. 2012). The AVE of NPL, ICO and RAB are 0.528, 0.520 and 0.552, respectively, which are above 0.50. The maximum shared variance (MSV) must be less than AVE for the factor validation (Almén et al. 2018; Byrne 2010; Hair et al. 2012) and Table 2 shows that MSV is less than AVE. Harman's one-factor test was used to find the common method bias or the common method variance (Harman 1961) to ensure the validity of all the extracted factors (Sharma et al. 2009). The percentage cumulative variance of the factor extracted from the explorative factor analysis of the variables is 34.15%, which is less than 50%, which reflects that there is no common method bias in extracting factors (Podsakoff et al. 2003). The above result shows that there are no issues of reliability and validity, making the entire construct reliable and valid, which can be used further for the structural model for the study of the effect of the constructs.

**Table 2.** Comparison of acceptable (standard) parameter values and the observed values of the latent variables of the model.

| Validity of Constructs of the Measurement Model | | | | | | | | | |
|---|---|---|---|---|---|---|---|---|---|
| Scale Items of the Constructs | Literature Sources | Factor Loading | CR | AVE | MSV | Max R(H) | NPL | ICO | RAB |
| **Neuroplasticity (NPL)** | | | 0.766 | 0.528 | 0.214 | 0.826 | 0.726 | | |
| NPL1 | (Syriopoulos and Bakos 2019), (Fiske 1993), (Haritha and Uchil 2020), (Bondia et al. 2019) | 0.878 | | | | | | | |
| NPL2 | | 0.649 | | | | | | | |
| NPL3 | | 0.625 | | | | | | | |
| **Investors' Cognition (ICO)** | | | 0.809 | 0.520 | 0.315 | 0.861 | 0.435 | 0.721 | |
| CC | (Hosaini and Saadatmand 2014), (Kumaravelu 2019), (Talwar et al. 2021), (Duong et al. 2014), (Sohaib et al. 2019 ) | 0.714 | | | | | | | |
| MC | | 0.892 | | | | | | | |
| HC | | 0.598 | | | | | | | |
| SC | | 0.646 | | | | | | | |
| **Risk Absorption (RAB)** | | | 0.786 | 0.552 | 0.315 | 0.795 | 0.463 | 0.561 | 0.743 |
| RI | (Robbins 2011), (Statman 2017), (David and Matu 2017), (Ferree and Merrill 2000), (Bhushan 2014) | 0.747 | | | | | | | |
| RS | | 0.798 | | | | | | | |
| SP | | 0.679 | | | | | | | |

CR—Composite Reliability, AVE—Average Variance Extracted, and MSV—Maximum Shared Variance.

The Chi-squared value or the CMIN/DF value is 1.790 (Schreiber et al. 2006; Almén et al. 2018). The Root-Mean-Square Residual (RMR) is 0.055 (Hu and Bentler 1999; Arbuckle 2007), the Goodness-of-Fit Index (GFI) is 0.907 (Almén et al. 2018; Schreiber et al. 2006), the Adjusted Goodness-of-Fit Index (AGFI) is 0.893 (Hu and Bentler 1999; Jöreskog and Sörbom 1993) and the Parsimony-Adjusted Goodness-of-Fit Index (PGFI) is 0.785 (Schreiber et al. 2006; Tanaka 1987). The Normed Fit Index (NFI) is 0.906, the Incremental Fit Index (IFI) is 0.956, the Tucker–Lewis Index (TLI) is 0.952 (Hu and Bentler 1999; Schreiber et al. 2006) and the Comparative Fit Index (CFI) is 0.956 (Hu and Bentler 1999; King et al. 2000). The Root-Mean-Square Error of Approximation (RMSEA) is 0.040 (Hu and Bentler 1999; Schreiber et al. 2006). These values are proven to be accepted and the close fit norms. All the experimental obtained values fit the parameter values. This means that the constructs taken for the study are reliable and valid, along with satisfying all the pre-requisites that are necessary for a relational proof modeling study.

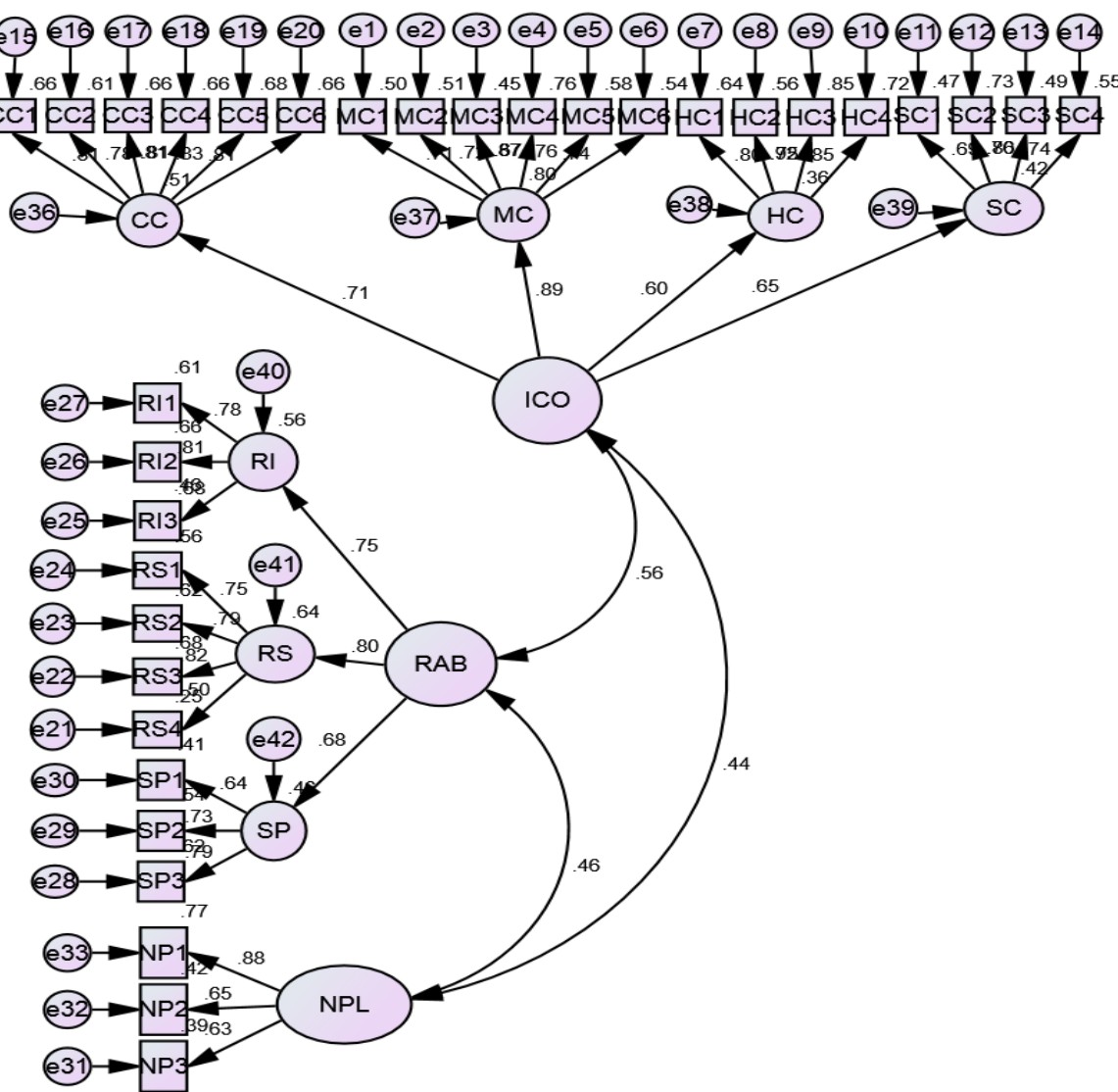

**Figure 1.** Measurement model of investors' cognition.

### 4.2. Neuroplasticity Structural Model

The structural model i.e., Figure 2 is drawn to prove the direct effect of investors' cognition and investors' risk absorption on neuroplasticity in investors, and the indirect effect of investor cognition on the neuroplasticity in investors mediated through risk absorption.

The Chi-squared value or the CMIN/DF value is 1.790 (Schreiber et al. 2006; Almén et al. 2018). The Root-Mean-Square Residual (RMR) is 0.055 (Hu and Bentler 1999; Arbuckle 2007), the Goodness-of-Fit Index (GFI) is 0.907 (Almén et al. 2018; Schreiber et al. 2006), the Adjusted Goodness-of-Fit Index (AGFI) is 0.893 (Hu and Bentler 1999; Jöreskog and Sörbom 1993) and the Parsimony-Adjusted Goodness-of-Fit Index (PGFI) is 0.785 (Schreiber et al. 2006; Tanaka 1987). The Normed Fit Index (NFI) is 0.906, the Incremental Fit Index (IFI) is 0.956, the Tucker–Lewis Index (TLI) is 0.952 (Hu and Bentler 1999; Schreiber et al. 2006) and the Comparative Fit Index (CFI) is 0.956 (Hu and Bentler 1999; King et al. 2000). The Root-Mean-Square Error of Approximation (RMSEA) is 0.040 (Hu and Bentler 1999; Schreiber et al. 2006). These values of the parameters are satisfying the accepted and the close fit norms. This, again, proves that the requisites parameters for the constructed variables are taken into consideration before model fitting and found to be correct, and hence the model can be proved.

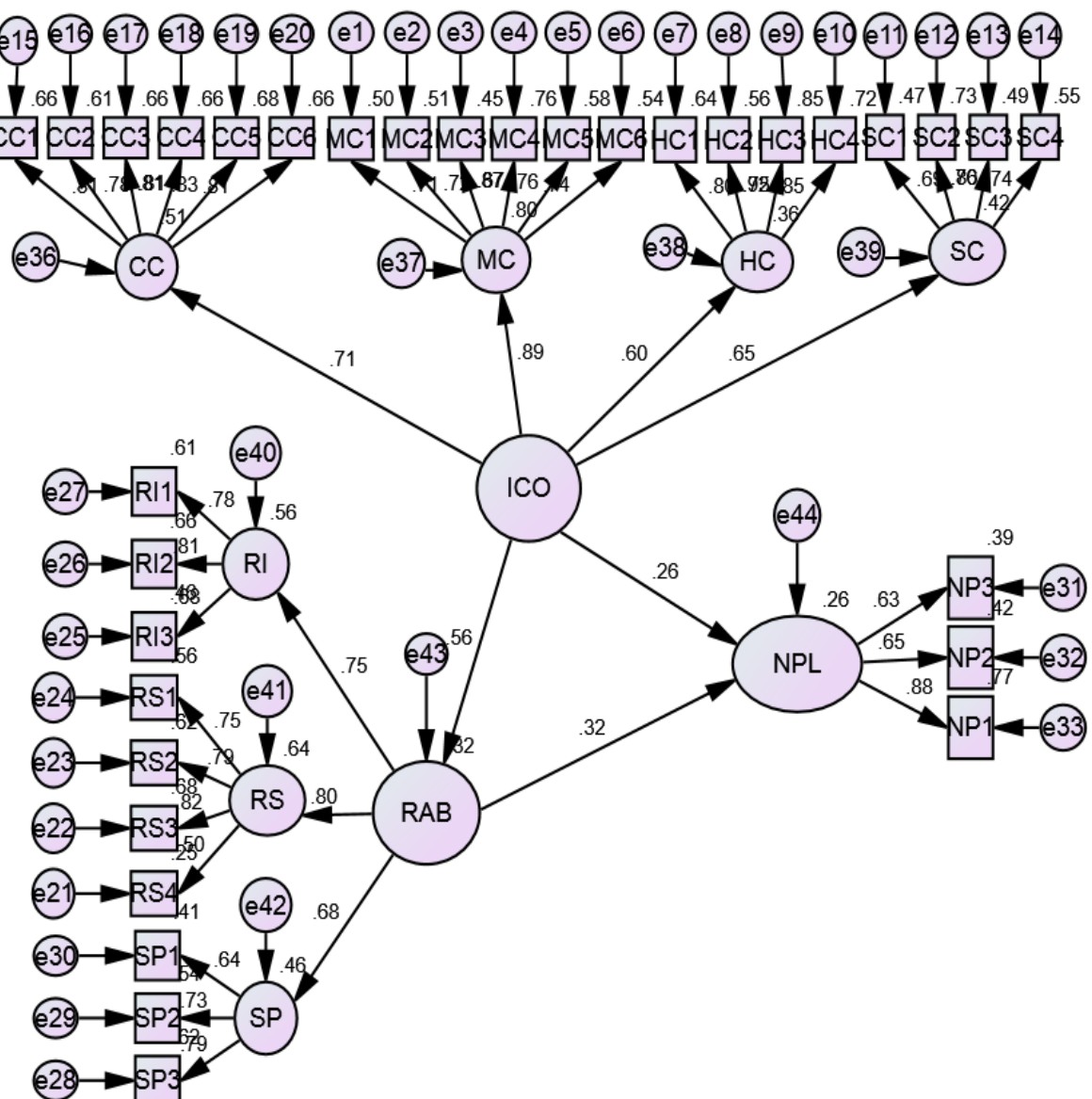

**Figure 2.** Mediating structural model.

The structural model, however, can be rejected, if it cannot comply with some of the parameter values. The rejection criterion for any structural model is the rejection combination explained below (Hu and Bentler 1999; Marsh et al. 2004). The rejection of a model fit can be conducted if any of the below-stated combinations are found.

1.  SRMR > 0.080 and RMSEA > 0.050;
2.  SRMR > 0.080 and RMSEA > 0.060 (sample size, case and model sensitivity);
3.  SRMR > 0.080 and CFI < 0.95;
4.  SRMR > 0.080 and CFI < 0.96 (sample size, case and model sensitivity).

The standardized Root-Mean-Square Residual (SRMR) is 0.0465, which is less than 0.080. The CFI is 0.956, which is more than 0.95 norms. The RMSEA is 0.040, which is less than 0.050. As the model does not undergo the rejection criteria, the model holds good. Therefore, all the relations in the structural model prove to be correct.

*4.3. The Testing of Hypotheses and the Discovery of the Effects through a Structural Model*

Table 3 shows the results of the tested Hypotheses 1–3.

**Table 3.** Relation among the latent variables (direct and mediating effects of independent variables on the dependent variable).

| | | | Description of Hypotheses Relating to the Structural Model | | | | | | | |
|---|---|---|---|---|---|---|---|---|---|---|
| Sl. No. | Hypotheses | | Dependence On | | Estimate | S.E. | C.R. | *P* | Standardized Regression Weight | Remarks on Hypotheses |
| 1 | H1 | NPL | <— | ICO | 0.248 | 0.116 | 3.907 | *** | 0.257 | Supported |
| 2 | H2 | NPL | <— | RAB | 0.356 | 0.181 | 3.032 | *** | 0.319 | Supported |
| 3 | H3 | RAB | <— | ICO | 0.486 | 0.076 | 6.943 | *** | 0.561 | Supported |

*** represents 0.000, which means *p* value or the significance value is 0.000.

Hypothesis 1 by approximately 26% (standardized regression weight is 0.257), as shown in the above table, so Hypothesis 1 (H1) is supported.

Hypothesis 2 by approximately 32% (standardized regression weight is 0.319), as shown in the above table, so Hypothesis 2 (H2) is supported.

Hypothesis 3 by approximately 56% (standardized regression weight is 0.561), as shown in the above table, so Hypothesis 1 (H1) is supported.

The standardized total effect of the different dimensions of investors' cognition (ICO) significantly affects the neuroplasticity of the investors (NPL) by approximately 44% (standardized regression weight is 0.435).

## 5. Discussion

The most important insight of this study is to discover the direct effect of cognitive dimensions on the neuroplasticity of psychologically injured investors due to their past experience with financial losses and the indirect effect of cognition in the presence of mediating factors, i.e., risks absorption. The other set objectives were to discover the effects of cognitive factors on the risk endurance of the risk-absorption nature of investors. Although many previous studies are in combinations related to cognition and decision making, and cognition and risk tolerance but few studies focus on the cognition that is helpful in the neuroplasticity of investors (Merkle 2011). This study found by taking neuroplasticity as the dependent variable, cognition dimensions as independent variables and risk absorption as the mediating variable, there is a significant and direct effect of cognitive dimensions on neuroplasticity and the partial mediating impact through risk absorption. The study also concluded that the risk-absorption attitude has a direct effect on the neuroplasticity of investors' psychology. The mediating effect of cognition through risk absorption proved to be more than a direct effect. The different dimensions of cognition, in combination and in isolation, also provide a significant effect on the dependent variables (Behera et al. 2021). The above inferences determine that the different dimensions of cognition, which are cold cognition, hot cognition, social cognition and metacognition, can help investors to find the strengths and weaknesses related to investment knowledge and avenue selection, so that the problems in investment decisions can be tackled and resolved. The four dimensions of cognition can bring in a sense of reasoning and understanding related to risk, return, speculation, fundamental information, and opinions, while making investment decisions and psychological enabled decisions in the future, and letting go or forgetting old loss experience. The present study also illustrates that the cognitive elements developed in an investor can reciprocate to the adherence of risk beyond the tolerance level, which can result in greater risk bearing in investments. The other objective of the current research was to ascertain the effect of risk-absorption behavior on neuroplasticity and found that risk-absorption behavior heals past psychological injury created due to dreadful loss and volatility related to investments in the equity market and can bring a better outlook towards investment scenarios (Wang and Deng 2018). Therefore, policymakers (persons and institutions related to the stock market, mutual funds and SIP organizations, and the regulators of the stock markets) should create strategies, considering the different brain

mechanisms arising from the cognitive processes of investors found in the present study, to deal with those investors who are reluctant to invest in the market due to their past, negative experiences. The different organizations involved in asset management need to attract more potential investors. The investors who have already invested in the stock market or risky assets can be potential investors. We can encourage them to make investments by understanding their cognitive abilities and emotionally driven attitudes. This study can help the policymakers of stock exchanges, brokers, asset management companies, and mutual fund agencies to understand the different kinds of potential investors, the variables driving their investment decisions and can use the different types of cognitive theories and accessed cognition sources (mainly information sources) to influence their future investment decision making. These information sources can be used as the basis and tools for advertisement, so that emotionally broken investors can understand, where they made mistakes in their earlier investments and how they can gain more profit in the future using their current knowledge and information.

## 6. Limitations and Future Research Avenues

The current study focused on investors in India using 506 respondents. More respondents from different South Asian regions may present better results. The analysis from subcontinent data may present a better understanding of the attitude of the investors towards past losses and position of psychological damage and the analysis can present a better picture of their reluctance towards investment. The present study focused on the cognitive dimensions. Other dimensions concerning attitude formation, such as conative, affective and psychometric, can be used for detailed study. More data analysis concerning sentiment analysis can present more relevant knowledge about the purchasing decisions of the subcontinent investors.

## 7. Conclusions

COVID-19 has already taken its toll on the Indian capital market by reducing the investors' faith in the market (Khan and Natu 2020), which ultimately reduced the flow of funds towards the market, directly and indirectly, through SIP and mutual funds (Gurbaxani and Gupte 2021). Understanding investors' behavior has been a predominant task, not just for the financial institutions, but for many policymakers (Kumari et al. 2020). It is not only sufficient to encourage new investors to invest in the equity market, but also to effectively address the issues faced by the experienced, but potential investors, those who are reluctant to invest due to past losses. From this perspective, strategies can be formulated with the help of financial product marketers (associated with stock market investments) by screening the findings of the current study, in which different dimensions and sources for the cognitive processing of information must be taken into consideration for making strategies, which can build confidence among past investors. The financial product markets can also show the investors where they were wrong in earlier investments, and how can that be tackled in future investments by developing their cognitive abilities. These dimensions of cognitions can help to forgo the memory of past losses and can build a risk-absorption attitude to further invest in financial equity market assets. This prolonged investment from different generations of investors can present the Indian capital market as more trustworthy and growing.

**Author Contributions:** Conceptualization, S.S.N. and Y.D.P.B.; Data curation, S.S.; Investigation, Y.D.P.B.; Methodology, T.R.S. and S.S. All authors have read and agreed to the published version of the manuscript.

**Funding:** The research received no external funding.

**Institutional Review Board Statement:** Not applicable.

**Informed Consent Statement:** Not applicable.

**Data Availability Statement:** The data are available from the author and can be produced upon request.

**Conflicts of Interest:** The authors declare no conflict of interest.

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
