# Peer review of "Examining Risk Absorption Capacity as a Mediating Factor in the Relationship between Cognition and Neuroplasticity in Investors in Investment Decision Making"

_ijfs, doi:10.3390/ijfs10010021_

Round 1
Reviewer 1 Report
English needs to be improved. It is very difficult to understand the results of the research because of the poor level of the language.
The acronyms need to be introduced when used for the first time. This comment applies also for the acronyms that are contained in the tables. Moreover, it would be appreciable the use of unambiguous acronyms: for example, neuroplasticity is identified with NP (page 5, line 209) and with NPL (pag 6, line 260).
Author Response
Comments: English needs to be improved. It is very difficult to understand the results of the research because of the poor level of the language.
The acronyms need to be introduced when used for the first time. This comment applies also for the acronyms that are contained in the tables. Moreover, it would be appreciable the use of unambiguous acronyms: for example, neuroplasticity is identified with NP (page 5, line 209) and with NPL (pag 6, line 260).
Compliance:
Thank you for the suggestions sir.
1- Unambiguous acronyms are now placed. The different acronyms are made clear.
2- The results show parameter value and the accepted values of measurement model and structural model justifying the correctness of the model. The reliability and validity of all the variables and factors taken for the model are explained in the result chapter are correct. Earlier I have only proved the model in the result part elaborated the significance in the discussion part, interpreting the model and its importance. However, with your suggestion, I have made the result chapter more understandable with interpretation.
Thank you again sir for the valuable comments.
Reviewer 2 Report
The authors have chosen a unique and interesting topic for their research. I would like to contribute my suggestions to improve the study, so I would like to suggest the following for the authors' consideration:
- The introduction of the study is very long, I suggest shortening it,
- I find the introduction structurally unintelligible. It contains a subsection 1.1, which has no counterpart in the text, but there are subsections underneath it, which should be placed somewhere,
- I propose to split the current introduction into an introduction and a literature review. The introduction should lay the groundwork for the topic, and the literature review should lay the foundations for the literature. The current sub-chapter 1.1 seems to me to be more of a literature review.
- The hypotheses should not be formulated in the literature review, but in the methodology section, and any methodological foundations should be placed there, with sufficient explanation. Also explain the software support,
- the results are eligible.
So far, I propose to include the following chapters in the article:
1. introduction
2. literature review
3. material and methodology
4. results
5. discussion
6. conclusions
I hope that my comments have helped to improve this paper.
Author Response
Comments: The authors have chosen a unique and interesting topic for their research. I would like to contribute my suggestions to improve the study, so I would like to suggest the following for the authors' consideration:
The introduction of the study is very long, I suggest shortening it,
- I find the introduction structurally unintelligible. It contains a subsection 1.1, which has no counterpart in the text, but there are subsections underneath it, which should be placed somewhere,
- I propose to split the current introduction into an introduction and a literature review. The introduction should lay the groundwork for the topic, and the literature review should lay the foundations for the literature. The current sub-chapter 1.1 seems to me to be more of a literature review.
- The hypotheses should not be formulated in the literature review, but in the methodology section, and any methodological foundations should be placed there, with sufficient explanation. Also explain the software support,
- the results are eligible.
So far, I propose to include the following chapters in the article:
1.introduction
2.literature review
- material and methodology
results- discussion
- conclusions
I hope that my comments have helped to improve this paper.
Compliance:
Thank you, sir for the suggestions. I have complied with each of your suggestions.
1- I have separated the introduction and literature review part.
2- I have made the chapterised as per the suggestions.
1.introduction
2.literature review
- material and methodology
results- discussion
- conclusions
3- The methodology is SEM (Structural equation modelling) showing the interrelation of the variables shown in the model through AMOS software.
4- Research hypothesis is taken in research methodology part.
Thank you again.
Reviewer 3 Report
The article cannot be published in its current version. There are too many doubts.
Comments:
- The term Neuroplasticity is misused in the article.
The human brain adapts to changing demands by altering its functional and structural properties (neuroplasticity) which results in learning and acquiring skills. Neuroplasticity is a medical term, and is not specified in the article.
- ‘Questionnaire was formed with 20 questions related to cognition, 10 questions related to risk absorption and 3 question related to neuroplasticity’ (232-233).
However, these questions are not known. It is difficult to comment on them. Of particular interest is the measurement of neuroplasticity (3 questions). The authors did not write anything about this.
- The authors specified hypotheses in two places (167, 208, 221 and 317-323), but differently!
- It is unclear how the authors research the relationship between cognition and neuroplasticity in investors.
- ‘The study outcomes are expected to aid in policy formulation as regards healing psychological trauma that potential investors suffered from losses in the past and overcoming reluctance to invest further in stock markets’ (26-28).
Are these recommendations for physicians?
- Figures 1 and 2 are not accurately described.
- The survey needs to be more clearly described. We don't know the questions in the questionnaire. So we don't know what was surveyed!
- ‘So, policy makers must make strategies viewing the different brain mechanism arising of cognitive process in investors found in the study to deal with those investors, who are reluctant to invest in the market due to their past experiences’ (365-368).
Who should create the strategies and how?
Author Response
- Comments: The term Neuroplasticityis misused in the article.
The human brain adapts to changing demands by altering its functional and structural properties (neuroplasticity) which results in learning and acquiring skills. Neuroplasticity is a medical term, and is not specified in the article.
Respected Sir, Thank you for precious comments.
Yes Sir, Neuroplasticity in the earliest study was basically from the field of science. It was initially considered as a process to reorganise the human injured brain. But it was notice that it was more of a psychological study that investigate the change in human brain due to access to new information, through learning and acquiring skills. It resulted in different kind of behaviour, which was not identified in one individual earlier but identified after the learning process (Neuroplasticity process). The term neuroplasticity is used metaphorically in the field of psychology and sociology to shape up human experiences and build new attitude towards certain stimuli.
Then some part of study relating to neuroplasticity shifted to psychology, where we can use learning ability in individual (Cognitive ability) to shape of their future behaviour.
In the prospect of Investment, the term neuroplasticity emphasized on reducing the effect of past losses, that brought negative perception towards stock market investment and can bring new attitude towards investment scenario with new and structured knowledge, which can be done by developing the cognitive ability and risk bearing characters. Some of the clarifications are presented in the below articles.
Neuroplasticity, also known as brain plasticity, is a term that refers to the brain's ability to change and adapt as a result of experience. (https://www.verywellmind.com/what-is-brain-plasticity-2794886)
Erkut, B., Kaya, T., Lehmann-Waffenschmidt, M., Mahendru, M., Sharma, G. D., & Srivastava, A. K. (2018). A fresh look on fi nancial decision-making from the plasticity perspective. International Journal of Ethics and Systems, 2(1), 1–11. https://doi.org/10.1108/IJOES-02-2018-0022
Njegovanović, A. (2018). Digital Financial Decision With A View of Neuroplasticity / Neurofinancy / Neural Networks. Financial Markets, Institutions and Risks, 2(4), 82–91. https://doi.org/10.21272/fmir.2(4).82-91.2018
Rugnetta, M. (2020). neuroplasticity | Different Types, Facts, & Research | Britannica. Britannica. https://www.britannica.com/science/neuroplasticity
- ‘Questionnaire was formed with 20 questions related to cognition, 10 questions related to risk absorption and 3 question related to neuroplasticity’ (232-233).
However, these questions are not known. It is difficult to comment on them. Of particular interest is the measurement of neuroplasticity (3 questions). The authors did not write anything about this.
Sorry sir, I could not place the questions earlier due to page bulkiness. The requirement of the software that was taken for analysis of the study is to place code instead of the whole question (Variable). But after your suggestion, I have placed the questions along with their codes in the article.
The authors specified hypotheses in two places (167, 208, 221 and 317-323), but differently!
I am really sorry sir, I have corrected it. Thank you so much for your guidance.
- It is unclear how the authors research the relationship between cognition and neuroplasticity in investors.
Sir, as this is deductive study, the relationship was constructed on the basis of literature reviews. We have found the relationship substantiated through literature in different fields like attitude formation, investment intention, purchase or investment decision etc. which are placed in the literature review heading and proved in the analysis and result heading.
- ‘The study outcomes are expected to aid in policy formulation as regards healing psychological trauma that potential investors suffered from losses in the past and overcoming reluctance to invest further in stock markets’ (26-28).
Are these recommendations for physicians?
Respected sir, I have mentioned now that strategies could be formulated by the financial product marketers, those who have involved in the stock market investment, to encourage the investors, whose thoughts and perceptions are affected by past losses.
- Figures 1 and 2 are not accurately described.
Sir, I have made modifications explaining the different figures and their uses in the study.
- The survey needs to be more clearly described. We don't know the questions in the questionnaire. So we don't know what was surveyed!
The questions are explained in the study now as per your instruction sir.
- ‘So, policy makers must make strategies viewing the different brain mechanism arising of cognitive process in investors found in the study to deal with those investors, who are reluctant to invest in the market due to their past experiences’ (365-368).
Who should create the strategies and how?
Institutions relating to stock market, mutual fund and SIP organisation and the regulators of the stock markets can use these knowledge by providing knowledge to different group of investors by applying different cognitive theories and can influence and encourage them to invest again in stock market. They can also show the investors, where they were wrong in earlier investment and how the problems can be tackled down in future scenarios by developing their cognitive abilities
Sir, I have added these lines to project a better relevance as per your instructions.
Reviewer 4 Report
The main aim of the paper is to investigates the mediating effect of risk absorption attitude in the relationship between cognition and neuroplasticity in investors. The methodology used is appropriate and the results are clearly demonstrated. The conclusion is supported by the research results.
In the discussion part authors should name the studies related to cognition and decision making and cognition and risk tolerance as their study is the extension to that area.
Author Response
Comments: The main aim of the paper is to investigates the mediating effect of risk absorption attitude in the relationship between cognition and neuroplasticity in investors. The methodology used is appropriate and the results are clearly demonstrated. The conclusion is supported by the research results.
In the discussion part authors should name the studies related to cognition and decision making and cognition and risk tolerance as their study is the extension to that area.
Compliance:
Thank you, sir for your precious comments. I have added little more clarification in the discussion part as per your instruction.
Round 2
Reviewer 1 Report
Despite the improvements, extensive editing of English language and style is necessary.
Author Response
Respected Sir,
I have made many changes by consulting with other authors and some professors in English. Corrections made to the article can be seen by track changes option in word.
Thanks and regards
Dr. Yadav Devi Prasad Behera
Reviewer 3 Report
Good luck!
Author Response
Thank you so much sir for your suggestions.
I have also corrected some grammatical mistakes, which can be seen through the track changes option.
This manuscript is a resubmission of an earlier submission. The following is a list of the peer review reports and author responses from that submission.